# Assessing the Sustainability of Long-Term Care Insurance Systems Based on a Policy–Population–Economy Complex System: The Case Study of China

**DOI:** 10.3390/ijerph19116554

**Published:** 2022-05-27

**Authors:** Rong Peng, Xueqin Deng, Yinghua Xia, Bei Wu

**Affiliations:** 1Institute of New Development, Guangdong University of Finance and Economics, Guangzhou 510320, China; dengxueqin@gdufe.edu.cn; 2Guangdong Provincial Institute of Public Health, Guangdong Provincial Center for Disease Control and Prevention, Guangzhou 511430, China; helen.xiayh@foxmail.com; 3Rory Meyers College of Nursing, New York University, New York, NY 10012, USA; bei.wu@nyu.edu

**Keywords:** long-term care insurance, policy modeling, policy strength, coupling coordination degree, pilot scheme

## Abstract

Although China launched long-term care insurance (LTCI) pilot program in 2016, there are great challenges associated with developing a sustainable LTCI system due to limited financial resources and a rapid increase in the aging population. This study constructed an LTCI policy–population–economics (PPE) system to assess the sustainability of the LTCI system in China. Based on the latest 76 LTCI policy documents published between 2016 and 2021, this study evaluated the strength of LTCI policy modeling in 14 pilot cities by constructing a policy modeling consistency (*PMC*) index containing 9 main variables and 36 sub-variables. The coupling coordination model was used to evaluate the interaction between LTCI policy, population aging, and economic development. The results showed that the *PMC* index ranged from 0.527 to 0.850. The policy strength of Qingdao, Nantong, and Shanghai was the highest (*PMC* > 0.8). Anqing, Qiqihaer, Chongqing, and Chengdu had the lowest level of policy strength (*PMC* < 0.6). The main policy weaknesses were the coverage of the LTCI, the sources of funds, the scope of care services, and benefit eligibility. The coupling coordination degree of PPE systems varied from 0.429 to 0.921, with a mean of 0.651. Shanghai, Nantong, and Suzhou had the highest level of coordination. The coordination between subsystems of PPE in most pilot cities (12 of 14 cities) was at a basic or low level. The findings from this study concluded that the coordination within the PPE system should be improved to develop a sustainable LTCI system. To improve the coordination of the PPE system, it is suggested that the country should maintain sustainable economic growth and modify LTCI policies based on demographic transitions and economic development.

## 1. Introduction

With the rapid growth of the aging population, the corresponding increase in long-term care (LTC) needs has posed great challenges to the sustainability of providing and financing care [1]. While the long-term care expenditure has kept growing over recent years, a large number of older adults with a disability lack general long-term care and financial support [2]. For example, the expenditure of LTC services reached 1.5% of gross domestic product (GDP) across OECD countries in 2019, 30–50% of older people with a disability reported that they did not receive sufficient LTC support [2]. In order to finance long-term care, some countries, such as Germany, Japan and South Korea, have introduced public long-term care insurance (LTCI) during recent decades. In China, the proportion of the population aged 65 or above rose to 13.5% in 2020 [3], and 52.71 million older adults have at least one limitation on daily life activity [4]. Since the majority of older adults with a disability are cared for by family members, they are accompanied by heavy family burden [5] and unmet long-term care [6].

To explore different methods of setting up a long-term care insurance (LTCI) system, China launched an LTCI pilot program in 15 cities in June 2016 [7]. Although China has become the second largest economy in the world in terms of GDP (USD 14.73 trillion in 2020, equivalent to 70% of that of the United States and accounting for 17.42% of the world), China’s GDP per capita was USD 11,300 in 2020, which is much lower than that of Germany in 1995 (USD 25,100), Japan in 2000 (USD 26,900), and Korea in 2008 (USD 27,464), when their respective LTCI systems were established. From the perspective of population aging trends, 13.5% of the population were over 65 years old in China according to the data of the seventh census in 2020. The proportion of the population over 65 years old was higher than that of Korea in 2008 (10.3%), close to that of Germany in 1995 (15.5%), and slightly lower than that of Japan in 2000 (17.4%) at the time of the establishment of their LTCI systems. Thus, the establishment of the LTCI system reflects the Chinese government’s willingness to actively deal with population aging.

It was reported that the LTCI pilot program in China had some impact after its implementation [8,9,10]. Using some pilot sites as examples, 752.8 thousand people in Chengdu received LTCI benefits amounting to yuan 843 million (equivalent to USD 132 million), which reduced the financial burden of families with disabled members by 44.31% [8]. The medical expenses of families with disabled members in Nantong were reduced from yuan 162 million (USD 25.5 million) to yuan 99.7 million (USD 15.7 million; about 38.5%) since the launch of the LTCI pilot [9]. Shanghai promoted the development of a long-term care industry by formulating action plans to encourage employment in this industry and trained 68 thousand nursing staff [10]. However, the Chinese central government has yet to establish a nationwide LTCI system but rather decided to expand the LTCI pilot sites to 14 other cities in China in 2020 [11]. At the current stage, China’s LTCI demonstration in the first 15 pilot cities has not accumulated sufficient evidence to formulate an appropriate policy scheme in line with the actual situation of the entire nation [12].

The major factors for the Chinese government to consider when establishing the LTCI are the economic development level and population aging [7,11]. There exists a complex interaction between LTCI policy, economic development, and population aging. On the one hand, the launch of LTCI increases the labor supply and stimulates the development of related industries. For instance, up to the end of 2020, the pilot city of Shanghai built up its LTC service supply market to include 750 nursing homes and 436 family-based care institutions [10]. The LTCI provides financial reimbursement to the persons receiving care in elder care institutions and residential care facilities. On the other hand, the economic development level determines the ability to finance the LTCI, which is related to the feasibility of the system [5,12]. Population aging is the main determinant of demand for LTCI and is related to the necessity of the system [7,11]. The interaction between the LTCI policy, economic development, and population aging trigger dynamic interactions in the internal structure, which is regarded as a coupling coordination mechanism.

Coupling refers to the measure by which two or more entities depend on each other, which originates from the testing and study of the interaction and correlation of multiple systems in physics [13]. Coordination can be used to describe the interaction between two or more systems that are not simply linearly correlated with each other [14]. The coupling coordination degree is a term that can be used to measure the coordination relationship [15]; it is a suitable instrument for analyzing the mutual effects between LTCI policy, economic development, and population aging. This study examined the coupling coordination between LTCI policy, economic development, and population aging to assess the sustainability of China’s LTCI system.

Compared with those of developed countries, China’s current social pension security system, pension service system, pension service facilities, and human and capital reserves are not well prepared to deal with the problem of rapid population aging under the condition of “getting old before getting rich” [16]. It was announced that an LTCI policy framework adapted to China’s economic development level and aging trend will be formed during the 14th Five Year Plan Period (2021–2025) [11]. One of the challenges now faced by China’s health care system is to determine the policy framework of an LTCI in the context of its current population aging and economic development levels. After all, there are huge differences between different regions in terms of economic development and population aging [17]. The central government has put forward guidelines with a large number of policy options to encourage local governments to carry out LTCI pilot programs. Assessing the sustainability of the LTCI system can not only help to develop a nationwide LTCI system but also provide insights for other countries facing similar challenges.

## 2. Literature Review

The primary concerns associated with an LTCI system are related to the future sustainability and affordability of long-term care financing and the equality of the current funding mechanism [1,2]. Government funding is increasingly burdened by the financing of long-term care, especially for the countries with a social LTCI system, such as in Germany and Japan [18,19]. Currently, some countries have carried out reforms on their LTCI systems to make them sustainable by increasing the individual payment or reducing the reimbursement rate to mitigate the government’s burden [18]. Generally speaking, how to keep the LTCI system sustainable in the context of population aging is a common problem worldwide [20,21].

Since China launched the LTCI pilot program, many studies have compared the policy documents issued by local governments to assess the implementation schemes of LTCI in pilot cities. They have found that the insurance coverage, eligibility, financing source, care service, benefits, and payments have much in common as well as some differences [22,23]. Several researchers have conducted empirical studies to evaluate the performance of the LTCI implementation in China. Zhang and Yu (2019) examined the willingness of the population to formally implement the LTCI policy in China [24]. Lei et al. (2022) found that older adults were less likely to report unmet activities of daily living needs in terms of care and that the intensity of informal care in pilot cities was reduced after the implementation of LTCI [25]. Feng et al. (2020) used medical insurance data to assess the effect of Shanghai’s LTCI pilot system on the residents’ medical expenses. It was found that China’s LTCI system had significantly reduced residents’ medical expenses [26]. Peng et al. (2021) used the system dynamics (SD) methodology to examine the impact of current and future LTC policies on the family care burden in China [27].

However, there are few quantitative evaluations on the strengths and weaknesses of the LTCI policy modeling of pilot cities in China. It is unclear which pilot city’s LTCI policy model is more valuable as a reference for China to establish a nationwide LTCI system that adapts to the current level of economic development and population aging. Since the purpose of the LTCI pilot is to establish a unique LTCI system suitable for China’s economic development and population aging trend, it is worthwhile evaluating the degree of coupling and coordination of the current LTCI policy model with economic development and population aging to identify key factors affecting the demographic and economic adaptability of the LTCI policy model.

The coupling coordination model has been widely used to evaluate the coordination between different systems. The major topic in the related research focuses on the coupling coordination between the economic, social, and environmental systems [28,29,30]. A few studies have explored the coupling coordination between health care institutions and citizens’ living standards [31], and between populations and urbanization settings [32]. To our knowledge, there is no research investigating LTCI policy strength and coupling coordination between LTCI policy, population aging, and economic development in China.

This study is an attempt to fill this gap through a quantitative evaluation of the LTCI policy modeling of the first pilot cities in China and an assessment of the sustainability of the LTCI system. This study makes the following contributions to the literature. First, it constructs an index system to estimate the strength of LTCI policy modeling. This index system helps to indicate the advantages of the LTCI policy schemes and is a tool that could be employed by other researchers. Second, this study evaluates the sustainability of the LTCI policy through the examination of the coupling coordination degree between local LTCI policy modeling, economic development, and population aging. It contributes to the literature by providing sustainability evaluations related to LTCI systems. Third, we discuss how to modify the policy in order to provide a comprehensive and sustainable LTCI system in China. As a developing country with a large aging population, China’s LTCI stimulates consumption and the emergence of the old-age economy [10]. China’s experiment provides a good case for exploring the establishment of the LTCI system.

## 3. Characteristics of China’s LTCI Pilot Scheme

The first 15 pilot cities are distributed in three regions of China: eastern coastal cities (Chengde, Shanghai, Nantong, Suzhou, Ningbo, Qingdao, and Guangzhou), inner China cities (Changchun, Qiqihar, Jingmen, Anqing, and Shangrao), and western cities (Chongqing, Chengdu, and Shihezi) (Figure 1). It can be seen that the pilot cities are mainly located in central and eastern China.

The pilot LTCI scheme mainly stipulates the coverage, financing mechanisms, and benefits packages. All the pilot LTCI programs cover participants who are participants of the Urban Employee Basic Medical Insurance. Some LTCI cities have expanded the insurance to cover participants who are participants of the Urban–Rural Resident Basic Medical Insurance. These cities are Nantong, Shangrao, Shihezi, Suzhou, Qingdao, Jingmen, and Shanghai. The main source of LTCI funding is medical insurance funds, followed by government subsidies, individual contributions, and employer contributions. Among them, Changchun, Shanghai, Guangzhou, and Ningbo rely entirely on medical insurance funds.

The benefit package, including care services and expense reimbursement, varies from city to city. The beneficiaries of all pilot cities include persons with severe disabilities, and they are extended to people with dementia in some cities (Shanghai, Nantong, Ningbo, Qingdao, and Chengdu). Most cities cover institutional care and home care (Changchun and Ningbo only cover institutional care). It is usually preferred to be reimbursed home care at a higher rate or a higher payment proportion. For example, the reimbursement rate of home care services is 90% in Shanghai, higher than that of institutional care services (85%). The services of all pilot cities include daily living care, and some cities also include preventive care, rehabilitation nursing, psychological counseling, medical care, and hospice care. All cities reimburse a fixed amount on a daily or monthly basis, with a maximum limit on the total number of hours or days. Some cities reimburse the beneficiary with a fixed proportion of the total expenditure. The policy schemes of pilot cities are listed in detail in Appendix A.

## 4. Materials and Methods

### 4.1. Study Design

#### 4.1.1. Data Collection

To describe and evaluate the LTCI modeling of pilot cities, we collected the LTCI policy documents issued by the General Office of the Chinese People’s Government and the Human Resources and Social Security Bureaus of the pilot cities. These documents are pilot guidelines and by-laws. They were downloaded from the websites of the central and local governments. The indicators of economic development and population aging were derived from the China Statistical Yearbook 2020, the China City Statistical Yearbook, and the Chinese National Economy and Social Development Statistics Bulletin.

#### 4.1.2. Participants

Due to a lack of data, one pilot city, Shihezi, was excluded. A total of 108 LTCI policy documents from the remaining 14 pilot cities were collected. After exclusion of the obsolete documents, 76 policy documents issued from 2016 to 2021 were analyzed.

### 4.2. Methods

#### 4.2.1. Text Analysis of Pilot Policies

All 76 policy documents were carefully reviewed. We made a comparison of the number of policy documents issued and the word count of documents between the pilot cities. Python was used to search for the top 10 high-frequency words in the policy texts.

#### 4.2.2. Policy Modeling Consistency (*PMC*) Index

*PMC* index (Ruiz Estrada, 2011) was used to evaluate the strength of the policy modeling of pilot cities [33]. The *PMC* index was measured using a multi-input–output table, in which the main variables and sub-variables were constructed to reflect the features of policy modeling. The *PMC* index enables policymakers and researchers to identify the level of consistency, as well as the strengths and weaknesses, of a policy model [33]. The *PMC* index has been widely used to evaluate policies due to its superiority over traditional models [34,35].

Table 1 shows the 9 main variables and 36 sub-variables of the *PMC* index. The 9 main variables that represent the main dimensions of LTCI policy design were: policy function (X_1_), policy scheme (X_2_), insurance coverage (X_3_), funding source (X_4_), care setting (X_5_), care service (X6), long-term care institution (X_7_), payment method (X_8_), and benefit eligibility (X_9_). Each main variable consists of several sub-variables. For example, the first main variable (policy function, X_1_) was made up of four sub-variables: system innovation (X_11_), managerial supervision (X_12_), standardized guidance (X_13_), and government support (X_14_). A multi-input–output table was established to measure the *PMC* index [33]. A binary system was used to assign a value to the sub-variable. If the sub-variable fit into the policy modeling, it was denoted by “1”, otherwise, it was denoted by “0”. For example, the sub-variable system innovation (X_11_) was scored as 1 if the term system innovation was included in the policy documents (see Appendix B).

The value of the main variable *X_t_* (*t* = 1,2,⋯,9) was obtained by dividing the sum of all sub-variables (of that particular main variable) by the total number of sub-variables.
$$X_{t}=\sum_{i=1}^{n_{t}}\frac{X_{ti}}{n_{t}};t=1,2,\cdots ,9;\;i=1,2,3\cdots$$
where *n_t_* is the number of sub-variables of the main variable *X_t_* and *X_ti_* is the value of sub-variable *i* of main variable *X_t_*, which equals 1 or 0. A smaller value of the main variable indicated that this was a weak point of policy modeling.

The value of the *PMC* index was obtained by dividing the sum of all the main variables by the total number of main variables.
$$PMC=\sum_{t}^{9}X_{t}/9$$

A higher *PMC* index indicated higher policy modeling strength.

#### 4.2.3. Coupling Coordination Degree

The coupling coordination degree that measures the complex interaction between LTCI policy, population aging, and economic development was used to assess the sustainability of the LTCI pilot system. Rapid population aging may lead to an increasing demand for long-term care [5]. It has become an important policy in many countries to finance long-term care [12,25]. Depending on the level of economic development, local governments implement different LTCI policies to fund long-term care systems, thereby improving their ability to cope with population aging. In this study, LTCI policy, population aging, and economic development were considered to be three interrelated subsystems that made up a complex system named the LTCI policy–population–economics (PPE) system.

The coupling degree of *n* subsystems can be computed using the following formula [28,31]:$$C={\left(\frac{\prod U_{i}}{{\left(\sum_{i}^{n}U_{i}/n\right)}^{n}}\right)}^{1/n}$$
where *n* denotes the number of subsystems and *U_i_* is the performance of subsystem *i*. *C* is the coupling degree, which represents the strength of interaction between multiple subsystems. The larger the coupling value, the more orderly the development direction between the subsystems and the more stable the relationship [32].

The coupling coordination degree reflects the extent to which multiple subsystems develop in harmony. It is calculated according to the following formula:
$$D=\sqrt{C*T};\;T=\sum_{i}^{n}\alpha_{i}*U_{i}$$
where *T* is the comprehensive development index measuring the overall effect of the subsystems. *α_i_* represents the weight of the contribution of the subsystem, given that ∑*α_i_* = 1. All the subsystems have equal weights. *D* refers to the coupling coordination degree. A higher value of *D* indicates a higher level of coherence between subsystems [32].

In accordance with the research purpose of this study, the LTCI policy strength measured by the *PMC* index was a unique evaluation indicator of LTCI policy subsystems (that is, *U_1_ = PMC*). Based on the literature review and index selection criteria, the evaluation index system of the population subsystem (*U*_2_) and economic subsystem (*U*_3_) were established. The population subsystem index, reflecting the level of population aging, included four variables: the proportion of the population aged 65 and above (*Y*_1_), the elderly dependency ratio (*Y*_2_), the child dependency ratio (*Y*_3_), and the natural population growth rate (*Y*_4_). The economic development indicator included four variables: GDP per capita (*Z*_1_), disposable income per capita (*Z*_2_), consumption expenditure per capita (*Z*_3_), and medical and health financial expenditure (*Z*_4_). The subscript *j* refers to city *j* (*j* = 1,2,⋯,14). *Y_ij_* and *Z_ij_* (*i* = 1, 2, 3, 4; *j* = 1,2,⋯,14) are the values of *Y_i_* and *Z_i_* for city *j*, respectively.

The performance or development level of the economic and population subsystems were calculated using the entropy method [36]. Information entropy reflects the disorder of the system, and it changes for each index [37]. The composite index of the population aging subsystem was calculated in three steps. First, the raw data *Y_ij_* was standardized using the following equation in order to eliminate the influence of dimension and magnitude. For positive values, $Y_{ij}^{\prime }=\frac{Y_{ij}-minY_{ij}}{maxY_{ij}-minY_{ij}}$; for negative values, $Y_{ij}^{\prime }=\frac{maxY_{ij}-Y_{ij}}{maxY_{ij}-minY_{ij}}$.

Second, the information entropy *E_i_* and the weight *W_i_* for the index *Y_i_* were calculated by the following formulas:$$E_{i}=-k\sum_{j}^{14}f_{ij}\ln\left(f_{ij}\right)\left(where,f_{ij}=\frac{Y_{ij}^{\prime }}{\sum_{j=1}^{14}Y_{ij}^{\prime }};k=\frac{1}{\ln14}\right),\mathrm{if}f_{ij}=0,\;f_{ij}\ln\left(f_{ij}\right)=0.$$
$$W_{i}=\frac{1-E_{i}}{4-\sum_{i=1}^{4}E_{i}}$$

Finally, the composite index *U*_2*j*_ of city *j* for subsystem *U*_2_ was calculated as follows:$$U_{2j}=\sum_{i=1}^{4}W_{i}Y_{ij}^{\prime }$$


The composite index *U*_3*j*_ of city *j* for subsystem *U*_3_ was calculated in the same way. Three coupling coordination degrees were calculated in this study: coupling coordination degree between LTCI policy and population aging, coupling coordination degree between LTCI policy and economic development, and coupling coordination degree between LTCI policy, population aging, and economic development.

The coupling coordination degree ranges from 0 to 1. Researchers often subjectively divide the coupling coordination degree into several levels [28,32]. In this study, the coupling coordination degree was divided into 0.9 or above (excellent), 0.8–0.9 (good), 0.6–0.8 (basic), 0.4–0.6 (low), and 0.4 or lower (no coordination) [32].

## 5. Results

### 5.1. Descriptions of Policy Documents

Table 2 shows the descriptive results of 76 valid policy documents. Shanghai issued the greatest number of policies (N = 19), followed by Qingdao (N = 17), Nantong (N = 14), and Suzhou (N = 10), while Chongqing issued the least number of policies (N = 2). Shanghai (N = 13) and Nantong (N = 12) had the largest number of valid documents. Shanghai’s policy documents were the most comprehensive, with a total of 50.4 thousand words in the 13 published documents. Chongqing’s policy documents were the least detailed, with 5.5 thousand words in two documents. The top 10 high-frequency words were long-term care, service, institution, insurance, evaluation, designated institutions, insurance coverage, disability, management, and funding (see Appendix C). This showed that the LTCI pilot policies were focused on factors related to care services, insurance coverage, and the financing of insurance.

### 5.2. PMC Index and Ranking

Table 3 presents the *PMC* index and ranking of the LTCI policy models in 14 pilot cities. It can be seen that the *PMC* indices ranged from 0.527 to 0.850. The range of *PMC* indexes was 0.323 and the dispersion coefficient was 0.157, which indicates that the *PMC* index varied greatly between the 14 cities. The three cities with the highest *PMC* indices (>0.8) were Qingdao (0.850), Nantong (0.838), and Shanghai (0.800); the lowest four (<0.6) were Anqing (0.593), Qiqihaer (0.568), Chongqing (0.567), and Changchun (0.527). The mean of the variables X_3_ (0.643), X_4_ (0.571), X_6_ (0.679), and X_9_ (0.679) was lower than total mean (0.733). The dispersion coefficients of X_4_, X_6_, and X_9_ were between 0.4 and 0.5, with X_3_ being above 0.5. This indicates a high LTCI policy weakness in terms of insurance coverage, funding sources, care services, and benefit eligibility.

### 5.3. Coupling Coordination Degree

The development level and coupling coordination degree of the LTCI PPE system in 14 pilot cities are presented in Table 4. The values of the coupling coordination degree ranged from 0.418 to 0.903. The mean of the coupling coordination degree was 0.636, indicating that the PPE system was at the basic coordination stage. The coupling coordination degree of Shanghai (0.903) was the highest and reached an excellent level. The overall evaluation indicators of subsystem, policy strength, level of economic development, and population aging of Shanghai were at a high level. Nantong reached a good coordination level (0.812). The other 12 pilot cities were at basic or low levels. Among them, the coupling coordination degree of Suzhou, Ningbo, Chongqing, Qingdao, Chengdu, and Guangzhou was at a basic level, ranging from 0.6 to 0.8. Jingmen, Changchun, Shangrao, Anqing, Chengde, and Qiqihaer were at a low coordination level, ranging from 0.4 to 0.6.

The coupling coordination degrees of LTCI policy and population of all pilot cities were greater than 0.8, indicating good or excellent coordination. However, the coupling coordination degree of LTCI policy and economy ranged from 0.4 to 1.0, showing a great variation in coordination.

## 6. Discussion

### 6.1. Common Characteristics of the LTCI Pilot Cities with a Higher Policy Strength

The findings from this study showed that the pilot cities with a high *PMC* index are usually located in eastern China, with more policy documents, a long pilot demonstration period, more comprehensive policy content, and updated policy documents. The three pilot cities with the highest *PMC* index (Qingdao, Nantong, and Shanghai) are economically developed cities in eastern China. A large number of policy documents with more details evidence of local government’s concern for the LTCI pilot.

Furthermore, the cities with a longer pilot demonstration period and significant progress in the policy pilots tended to have a higher *PMC* index. For example, Qingdao was the first city in China to officially launch an LTCI in 2012. This insurance plan initially provided long-term care with a focus on the medical care and then evolved to provide a professional long-term care and geriatric services for those with substantial long-term care needs. Currently, the Qingdao program has the most comprehensive coverage and benefits package among the pilot cities [22,38]. Therefore, it is not surprising that Qingdao has the most powerful LTCI policy strength.

Nantong and Shanghai, ranked second and third, are similar to Qingdao. They started the LTCI pilots early and have made at least one revision to their policies. Nantong launched its LTCI program in October 2015 and updated the LTCI supporting policies from 2016 to 2019. So far, Nantong has issued the largest number of policy documents, with nine main variables above average. Shanghai released its first pilot policy in December 2016 and adjusted and improved it in December 2017. The eight main variables of all the main variables were above or equal to the average, except for the source of fund (X_4_) which was below the average.

### 6.2. Main Weaknesses of LTCI Policy Modeling

This study found that insurance coverage, financing source, care service, and benefit eligibility were the main weaknesses of the LTCI policy modeling in China’s pilot cities. These policy deficiencies exposed the controversial issues surrounding the LTCI pilot in China. Some cities believe that the coverage of the LTCI should be extended to the entire population to reflect the fairness of the social insurance system [39]. However, others believe that only the older adults with a high risk of disability should be included. The extended inclusion of all people in the long-term care program will result in an increase in social security payments and an unfair financial burden [40]. In terms of the fund sources, some cities adopted the medical insurance fund as the only source of financing, while others raised funds from multiple sources, including government revenue, individual payments, and employer payments. Moreover, the payment standards, provisions for care services, and payment methods varied between the pilot cities. Therefore, the pilot program should be expanded to more cities and the weaknesses of the LTCI policy should be identified so that a national LTCI system can be established.

### 6.3. Influencing Aspects on Coupling Coordination Degree

This study found that the coupling coordination degree of the LTC policy–population–economics (PPE) system was at a basic coordination level. Only one city, Shanghai, reached the level of excellent coordination. Most pilot cities were at a low or basic level. The possible explanation for the low coupling coordination degree was that the LTCI policy was ahead of the economic development (the economic development lags behind). The pilot cities with the lowest coupling coordination degree (less than 0.5), such as Shangrao, Anqing, Chengde, and Qiqihaer, not only had a low policy strength but also had extremely low level of economic development. The economy’s lagging behind the LTCI policy strength may lead to unsustainable LTC financing in these cities [12].

The pilot cities with a basic coordination of the PPE system usually had the characteristics of a high policy strength and a medium level of economic development or alternatively a high policy strength and a low level of population aging. For example, Qingdao had the highest level of policy strength among the 14 cities but ranked number 6 in terms of the coupling coordination degree. The possible explanation was that its economic development was not enough to support its policy strength. Suzhou and Guangzhou were in a leading position in terms of the economic development among the 14 pilot cities. However, the population aging degree of the two cities was relatively low due to the young population age structure, indicating that the LTCI policy strengths of Suzhou and Guangzhou were ahead of their population aging level. Although the LTCI policy strength ahead of the economic development would help to deal with population aging in advance, it may lead to a waste of care service resources at present.

### 6.4. Policy Implications

This study revealed the weaknesses of the LTCI policy modeling in China’s first pilot cities. The LTCI policy in China is far from being perfectly coordinated with its demographics and economic development. In the future, the LTCI policies need to be modified based on the demographic transitions and economic development. The policy implications from the findings were helpful not only for a further development of LTCI policies in China but also for constructing the LTCI systems in other developing countries with similar population aging structures.

First, in order to establish a nationwide LTCI policy framework, in view of the identified issues this study found in the LTCI pilots, China needs to focus on the three aspects. Should China establish an LTCI system for all people or for some groups? How can we establish a fair and sustainable financing source for the LTCI? How do we define the eligibility criteria of the insurance benefits? These issues should also be considered for other countries with aging populations that are planning to establish LTCI systems. If a private LTCI system is adopted, only a small portion of the population can afford the insurance. If a universal public LTCI is adopted, there is a need to further improve the equity and accessibility of the LTCI system [41,42]. Therefore, a “low-level and broad coverage” LTCI system, which is similar to the current urban and rural medical insurance system in China [43], could be affordable for the developing economies. In addition, the scope of care services needs to be improved to meet the long-term care demands of the populations with special needs, such as the persons with dementia.

Second, China’s method is a referenceable way to pilot the LTCI system before the formal implementation, which helps to identify a sustainable policy modeling to be expanded to more sites. China’s pilot experiences showed that Shanghai’s policy modeling provides some good examples for the promotion of the LTCI system nationwide in China. Shanghai is the only pilot city that has both high policy strength and excellent coordination. This shows that the design of Shanghai’s LTCI scheme is in line with its local economic strength and population structure. Shanghai has the highest proportion of the elderly inhabitants among the economically developed cities in China. The city started the LTCI system pilot early and its LTCI pilot system has significantly reduced residents’ medical expenses [26]. Therefore, the LTCI policy modeling of Shanghai needs to be further studied so that the experience of Shanghai can provide good policy implications for other cities and regions.

Third, to build an LTCI system suitable for the demographic and economic development, it is necessary to improve the coordination of the PPE system. How to keep the financing sustainable is a dilemma in many countries with a rapid aging [1,5,21]. The high degree of coordination of the PPE system means a good matching between the LTCI policy, population aging and economic development. In order to make the LTCI system sustainable, it is important to maintain a sustained economic growth to provide a financial support for the LTCI. In addition, the governments should determine the level of funding for the LTCI based on the local economic development. If the LTCI fund relies too much on the government subsidies, this could potentially hinder future economic development and also give rise to an intergenerational inequality [44,45]. In China, the current source of funds in the pilot sites mainly comes from the allocation of the medical insurance funds and the financial subsidies, which makes the program subject to the financing level and financial support level from the medical insurance funds and this can lead to a risk of fund insufficiency [12,23,25]. A set of subsidy policies with dynamic adjustment mechanisms should be formulated in order to monitor the subsidy levels according to the fiscal revenue and expenditure so as to ensure the sustainability of the increased demand for funding.

## 7. Conclusions

This was the first study to evaluate LTCI policies’ strengths and their coordination with economic development and population aging among China’s pilot cities. The findings indicate that the insurance coverage, financing mechanisms, and benefits packages were main weaknesses of LTCI policy modeling. The findings from our study showed that the demographic and economic adaptability of the LTCI policies in China were far below where they need to be. Improving the coordination of the PPE system was critical to develop a sustainable LTCI system. To improve the coordination of the PPE system, it is suggested that the country should maintain sustained economic growth and modify the LTCI policies based on the demographic transitions and economic development.

## Figures and Tables

**Figure 1 ijerph-19-06554-f001:**
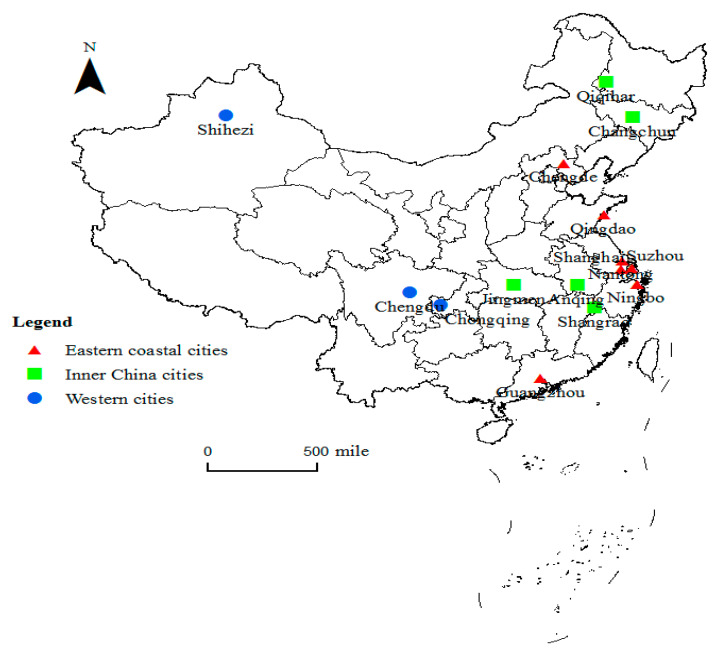
Locations of the first 15 pilot cities of the LTCI in China.

**Table 1 ijerph-19-06554-t001:** Main and sub-variables for the *PMC* index model.

Main Variables		Sub-Variables		Sub-Variables	
X_1_	Policy function	X_11_ X_13_	System innovation Standardized guidance	X_12_ X_14_	Managerial supervision Government support
X_2_	Policy scheme	X_21_ X_23_	Well-founded Clear objective	X_22_ X_24_	Clear responsibility Personnel training
X_3_	Population coverage	X_31_ X_33_	Urban employees Urban and rural residents	X_32_	Urban residents
X_4_	Funding source	X_41_ X_43_	Medical insurance fund Financial subsidy	X_42_ X_44_	Individual payment Employer’s subsidy
X_5_	Care setting	X_51_ X_53_	Health care facility Community	X_52_ X_54_	Residential care facility Home
X_6_	Care service	X_61_ X_63_ X_65_	Medical care Preventive care Psychological counseling	X_62_ X_64_ X_66_	Daily living care Rehabilitation care Hospice care
X_7_	Long-term care institution	X_71_ X_73_ X_75_	Hospital Geriatric hospital Daycare facility	X_72_ X_74_	Eldercare facilities Community health service center
X_8_	Payment method	X_81_ X_83_	Fixed payment Paid daily	X_82_ X_84_	Paid proportionally from the fund Paid monthly
X_9_	Benefit eligibility	X_91_	Disability	X_92_	Dementia

**Table 2 ijerph-19-06554-t002:** Descriptions of the local economy, aging population, and policy documents of 14 pilot cities in 2021.

City	GDP Per Capita (USD)	The Proportion of Older Adults Aged 65+	Policy Documents			
			Total Number of Documents	Number of Valid Documents	Number of Invalid Documents	Number of Words
Anqing	7960.35	13.73	5	4	1	13,217
Changchun	12,105.00	12.63	4	4	0	14,456
Chende	6462.59	11.73	3	1	2	11,789
Chengdu	16,272.96	14.27	5	4	1	26,825
Chongqing	11,935.33	15.34	2	2	0	6510
Guangzhou	24,621.61	12.57	7	2	5	51,012
Jingmen	11,049.95	13.07	5	4	1	18,663
Nantong	20,182.39	24.39	14	12	2	40,771
Ningbo	22,532.91	16.79	6	6	0	24,391
Qingdao	19,561.99	13.15	17	5	12	64,568
Qiqihaer	3567.79	8.14	5	4	1	21,430
Shanghai	24,755.71	17.86	19	13	6	50,440
Shangrao	5798.46	11.05	6	6	0	26,844
Suzhou	12,293.10	17.70	10	9	1	30,513
Total	—	—	108	76	32	401,429

Note: For brevity’s sake, the population aging indicators of the elderly dependency ratio, child dependency ratio, and natural population growth rate and the economic development indicators of disposable income per capita, consumption expenditure per capita, and medical and health financial expenditure are not shown in this table.

**Table 3 ijerph-19-06554-t003:** *PMC* index and ranking of LTCI policy models in 14 pilot cities.

City	X_1_	X_2_	X_3_	X_4_	X_5_	X_6_	X_7_	X_8_	X_9_	*PMC* Index	Ranking
Anqing	1.000	0.500	0.333	0.750	0.750	0.500	0.600	1.000	0.500	0.593	11
Changchun	0.750	0.750	0.667	0.250	0.500	0.500	0.600	0.750	0.500	0.527	14
Chengde	0.750	0.750	0.333	0.750	0.750	0.500	0.800	1.000	0.500	0.613	8
Chengdu	0.750	1.000	0.333	0.750	0.750	0.333	0.800	0.750	1.000	0.647	6
Chongqing	0.250	1.000	0.333	0.500	1.000	0.333	1.000	0.750	0.500	0.567	13
Guangzhou	1.000	0.500	0.333	0.250	1.000	0.833	0.600	1.000	0.500	0.602	9
Jingmen	0.250	1.000	1.000	0.750	0.750	1.000	0.800	1.000	0.500	0.705	4
Nantong	1.000	1.000	1.000	0.750	1.000	0.833	0.800	1.000	1.000	0.838	2
Ningbo	1.000	1.000	0.333	0.250	0.500	0.833	0.600	0.500	1.000	0.602	10
Qingdao	0.750	1.000	1.000	0.750	1.000	1.000	1.000	1.000	1.000	0.850	1
Qiqihaer	0.750	1.000	0.333	0.500	0.750	0.500	0.600	0.750	0.500	0.568	12
Shanghai	1.000	0.750	1.000	0.250	1.000	1.000	1.000	1.000	1.000	0.800	3
Shangrao	0.500	1.000	1.000	1.000	0.750	0.500	0.600	0.500	0.500	0.635	7
Suzhou	1.000	0.750	1.000	0.500	1.000	0.833	0.800	0.500	0.500	0.688	5
Mean	0.768	0.857	0.643	0.571	0.821	0.679	0.757	0.821	0.679		
Dispersion coefficient	0.349	0.220	0.517	0.435	0.221	0.366	0.212	0.251	0.366		

**Table 4 ijerph-19-06554-t004:** Development level and coupling coordination degree of the LTCI PPE system in 14 pilot cities.

City	PPE Subsystem Development Level			Coupling Coordination Degree				
	Policy Subsystem	Population Subsystem	Economy Subsystem	Between Policy and Population	Between Policy and Economy	PPE System	Ranking of PPE System	Coordination Level of PPE System *
Anqing	0.593	0.344	0.052	0.964	0.546	0.469	12	Low
Changchun	0.527	0.402	0.153	0.991	0.835	0.565	10	Low
Chengde	0.613	0.310	0.037	0.945	0.461	0.437	13	Low
Chengdu	0.647	0.450	0.354	0.984	0.956	0.685	7	Basic
Chongqing	0.567	0.545	0.384	1.000	0.981	0.701	5	Basic
Guangzhou	0.602	0.209	0.719	0.875	0.996	0.670	8	Basic
Jingmen	0.705	0.399	0.132	0.961	0.729	0.578	9	Low
Nantong	0.838	0.990	0.345	0.997	0.909	0.812	2	Good
Ningbo	0.602	0.503	0.543	0.996	0.999	0.740	4	Basic
Qingdao	0.850	0.334	0.408	0.900	0.936	0.698	6	Basic
Qiqihaer	0.568	0.322	0.029	0.961	0.430	0.418	14	Low
Shanghai	0.800	0.693	0.976	0.997	0.995	0.903	1	Excellent
Shangrao	0.635	0.196	0.097	0.849	0.677	0.479	11	Low
Suzhou	0.688	0.398	0.620	0.964	0.999	0.744	3	Basic

* Excellent: 0.9 or above; good: 0.8–0.9; basic: 0.6–0.8; low: 0.4–0.6; no: 0.4 or lower.

## Data Availability

The data presented in this study are available on request from the corresponding author.

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
