# Peer review of "Assessing the Sustainability of Long-Term Care Insurance Systems Based on a Policy–Population–Economy Complex System: The Case Study of China"

_ijerph, 2022, doi:10.3390/ijerph19116554_

Round 1
Reviewer 1 Report
I dare to make few comments for improvement, perhaps in the introduction I do miss a section explaining the model of care for the aging in that country. Since it is an international journal, I believe that a section expanding this information is more than necessary.
Another doubt that arises when I read the results, if different policies are found in different Chinese cities, how is the management of care for the elderly or dependency in China? Are the competences of this management transferred to the different local governments?
Reviewer 2 Report
Dear authors,
Thank you for great effort contributed into this manuscript.
Please find my comments below
- The LTCI was launched in 2016, but documents in 2015 were included. Please provide rationale and timeline.
- Please use the key term consistently, -population aging vs aging population. These two term are different in meaning.
- Please provide a full term of PCM before using the abbreviation
- In the abstract, you provide the highest level of PMC, it'd be better to present the cities with the lowest level as well.
- In abstract conclusion, you can add more merit by offering the strategies/means to improve PPE to reach the sustainability
- Methods section, there are some missing parts- study design, participants and sampling, setting. I understand that you already provide information somewhere in this manuscript; however, to report a good research paper we should follow the standard structure
- This paper provides a great benefit for practice and further research, well done. Nonetheless, English and presentation need to revise to suit the academic standard.
Best of luck
Reviewer 3 Report
This is an interesting paper, however, a bit overfocused on China's specific features of the care insurance system. It is a case study only, it concentrates on one particular program in one particular country. The author fails to demonstrate the overall relevance of the issue. To my mind, the scope of the paper should be expanded to increase the value of contributions to the literature. The author should emphasize the relevance of studying the sustainability-related aspects of insurance systems for developing countries (or emerging markets, whatever the author classifies) and only after summarizing the overall gaps, proceed with the specific situation in China. The gaps identified in the literature review are specific to one country. Do they differ in other countries? If yes, what are the differences, and how could they be interpreted? Also, if yes, what is the specific value of studying China? Therefore, the author should demonstrate a broader picture. The broader implications of the findings should be demonstrated in the discussion, with more emphasis on the out-of-China implications of the findings in other markets. The Conclusion must reflect this broader content of the paper, by emphasizing the novelty of the study, the potential implications of the results for academic literature (not only Chinese) and policies (developing markets, not only China)
Round 2
Reviewer 3 Report
My recommendations are now addressed. However, I recommend the manuscript go through a language check to improve the quality the language and style
Author Response
Point 1:My recommendations are now addressed. However, I recommend the manuscript go through a language check to improve the quality the language and style.
Response: Thanks for the reviewer’s comments. This manuscript has previously undergone English language editing by MDPI. Once again, we checked the grammar and spelling carefully and made some changes in the revised manuscript. We hope the English language can meet the journal’s requirement.